# East Africa's Policy and Stakeholder Integration of Informal Operators in Electric Mobility Transitions—Kigali, Nairobi, Kisumu and Dar es Salaam

**Jakub Galuszka** [1,*], **Emilie Martin** [2], **Alphonse Nkurunziza** [3], **Judith Achieng' Oginga** [4], **Jacqueline Senyagwa** [2], **Edmund Teko** [2] and **Oliver Lah** [1,2,5,*]

1    Habitat Unit, Technische Universität Berlin, 10623 Berlin, Germany
2    UEMI Africa Living Labs, Urban Electric Mobility Initiative (UEMI), 10437 Berlin, Germany;
     emilie.martin@uemi.net (E.M.); jacqueline.senyagwa@uemi.net (J.S.); edmund.teko@uemi.net (E.T.)
3    Centre of Excellence in Transport Planning, Engineering and Logistics, College of Science and Technology,
     University of Rwanda, P.O. Box 3900, Kigali, Rwanda; nzizaalphonse@gmail.com
4    Spatial Planning Department, Swedish School of Planning, Blekinge Institute of Technology,
     371 41 Karlskrona, Sweden; judithachieng.oginga@bth.se
5    Mobility and International Cooperation, Wuppertal Institute for Climate, Environment and Energy,
     10178 Berlin, Germany
*    Correspondence: jakub.galuszka@tu-berlin.de (J.G.); oliver.lah@wupperinst.org (O.L.)

**Abstract:** Electric mobility is beginning to enter East African cities. This paper aims to investigate what policy-level solutions and stakeholder constellations are established in the context of electric mobility (e-mobility) in Dar es Salaam, Kigali, Kisumu and Nairobi and in which ways they attempt to tackle the implementation of electric mobility solutions. The study employs two key methods including content analysis of policy and programmatic documents and interviews based on a purposive sampling approach with stakeholders involved in mobility transitions. The study findings point out that in spite of the growing number of policies (specifically in Rwanda and Kenya) and on-the-ground developments, a set of financial and technical barriers persists. These include high upfront investment costs in vehicles and infrastructure, as well as perceived lack of competitiveness with fossil fuel vehicles that constrain the uptake of e-mobility initiatives. The study further indicates that transport operators and their representative associations are less recognized as major players in the transition, far behind new e-mobility players (start-ups) and public authorities. This study concludes by identifying current gaps that need to be tackled by policymakers and stakeholders in order to implement inclusive electric mobility in East African cities, considering modalities that include transport providers and address their financial constraints.

**Keywords:** electric mobility; paratransit; informality; transportation; East Africa

## 1. Introduction

Facing climate change and associated issues such as humanitarian crises and growing socio-economic disparities, the global agendas uniformly recognize the urgent need to transition to sustainability. Development of low-carbon transportation in urban centers is positioned as a key element of this process, since the transportation sector is estimated by the Intergovernmental Panel on Climate Change (IPCC) to generate 23% of global energy-related greenhouse gas emissions [1]. Consequently, promotion of public transportation and electric mobility is identified as a crucial point in the combat against climate change [2]. Pathways to reach more sustainable urban mobility are structured around the three-pronged "Avoid-Shift-Improve" paradigm that promotes trip reductions and shifts towards public transport and non-motorized modes, together with increased vehicle efficiency [3]. Electrification of mobility is commonly conceptualized as part of the "Improve" component and as a low-carbon transport strategy [4]. This transition also links

with the necessity to phase out the old solutions [5] such as internal combustion engines (ICE), inevitably resulting in socio-economic consequences for those who depend on these technologies as a source of income and livelihood.

A specific focus on aspirational, quantitative improvements is visible in the major international development agendas while focusing, to a smaller degree, on the local complexities of the transition to a sustainability process. For instance, some authors argue that documents such as the New Urban Agenda reinforce path dependency on techno-managerial approaches and the ecological transformation paradigm, even though those have failed to work in the past [6]. Similarly, the Sustainable Development Goals (SDGs) are discussed to require stronger integration in local contexts and their evaluation measures to integrate more robust qualitative components [7]. At a broader level, even some of the most acknowledged eco-friendly planning solutions, such as promotion of a compact city design, walkability and public transportation, may have adverse social effects and lead to displacement or marginalization of low-income groups [8,9]. While the current pandemic may accelerate the transition to some of these solutions, it also puts an enormous economic strain on sectors operating outside of regularized and formal markets. The restrictive measures, enforcing reductions in mobility in the poorest parts of densely populated cities, epitomize the paradox embedded in the process of transitioning into sustainability. The costs of changes, which are beneficial for the majority of society, may be borne disproportionately by specific sectors or groups or affect them in a more direct manner.

Along the theoretical debate, ground realities of urban transformation illustrate similar difficulties in the prioritization of public transportation and e-mobility solutions, with East African cities also experiencing challenges when implementing integrated public transport solutions or pushing for new technical standards and fuel taxation [10].

Electric vehicles are beginning to enter the market in East Africa, mainly in the context of micro-mobility [11]. The situation is slowly evolving with the introduction of some electric vehicles, albeit at stages of small fleets or pilot projects. Such vehicles include two- and three-wheelers, including bikes, motorcycles and tricycles, as well as buses and cars, the latter both in the forms of private vehicles and for commercial purposes as taxi services. The transition towards electrification is a complex process, characterized by a triple value chain of vehicles, a charging and surrounding network [12], commonly requiring deep multi-sectoral cooperation, and seeing the emergence of new electric mobility players such as private transportation or energy companies. Therefore, e-mobility may add further complexities to local systems and, arguably, could run the risk of being contested on various levels.

Contestations as a reaction to transportation regulation and innovation are not uncommon. Recently, protests and unrest instigated by policies addressing climate change have been significant and concerned with different dimensions of transportation planning. This includes contestation of regulations on fuel taxation as in France during the "Yellow Vests" protests, the imposed technological standards in the electrification of jeepneys in Metro Manila [13] or Cape Town, where the extension of the bus system to townships was contested by operators of minibuses. Those examples illustrate why neglecting socio-economic issues in climate change-oriented policy planning may result in a spontaneous contestation of a specific solution and, in some cases, it being entangled with political agendas. More recently, these kinds of tensions have been fueled by the public health regulations linked to coronavirus disease-19 (COVID-19) and were perceived as a further threat to the economic wellbeing of small-scale entrepreneurs. The pandemic context is thus likely to further complicate the transitioning process to electro-mobility due to the disruption of supply chains and redirection of budgetary sources to other priorities.

For these reasons, addressing the issues of social justice [14], informal solutions, workers' rights [15] and similar issues emerges as a central component of transitioning processes (The term 'informal transport' is problematic in itself and it raises both negative and positive connotations depending on the field of study and represented sector. While we acknowledge those diversified perceptions we use it based on the prominence in literature

and apply interchangeably with terms such as: paratransit, operators, transport providers. At the same we acknowledge that in none of the cases operators are fully 'formal' or fully 'informal'). The social and economic aspect of integration of e-mobility solutions into local contexts should be considered as equally important as their technological appropriateness or regulatory mechanisms they enforce. The picture gets further complicated considering that transitioning to e-mobility happens within a variety of transport modes and geographies having distinct operational, financial and organizational patterns and may experience various reactions from operators and users. In some cases, transport transformation processes may be supported through bottom-up activism of the informal sector, local start-ups supporting mainstreaming of specific technology at a low cost or pressure from users who appreciate a specific mode (for instance, public opposition against a ban on moto-taxis in the center of Kigali) [10]. In others, the magnitude of required investments puts different challenges on the sector and arguably leads to various responses of the involved stakeholders. Urban and transportation studies have increasingly recognized this through a reflection on the interwoven relationship between formal and informal practice [16] and the institutional bricolage it creates [17]. Facing inherent conflicting rationalities in urban development contexts [18], development of policies which may support integration of various systems, for instance, through practices of co-production [19] or hybrid systems, could be positioned as a central step in increasing the acceptance of e-mobility and technological innovations.

The aim of the study is to investigate the policy level solutions and stakeholder constellations established in the context of e-mobility in East Africa and to specifically reflect on the positioning of informal operators in the e-mobility transitions.

More specifically, we inquire the following:

- How do the country- and city-level policies in the selected contexts tackle e-mobility transitions?
- Which stakeholders take an active role in the mainstreaming of e-mobility and what roles are those?
- What is the positioning of informal/semi-formal operators in the e-mobility transitions?

Specific attention is given to the issues of integration of existing transport service providers, who operate at the intersection of formal and informal transportation systems. It reflects on the recognition of diversified transport modes within the policies, inclusion of a different range of stakeholders into the transitioning process and differentiations in the focus of these policies, including environmental, economic, technical and social issues. At the same time, the first responses to those external regulations and formats by the private sector entrepreneurs and informal (semi-formal) actors are discussed. The article finalizes with recommendations on how the public sector can promote e-mobility solutions, through concrete actions and policies, while ensuring that negative socio-economic impacts of the transitioning process are mitigated. The analysis is conducted in the context of Kigali, Nairobi, Kisumu and Dar es Salaam.

## 2. Materials and Methods

This study takes an exploratory character and concentrates on the investigation of policy instruments and perceptions related to the introduction of e-mobility in mass transit in cities of East Africa. Two key methods are applied in the study including content analysis of policy and programmatic documents, and interviews with stakeholders involved in mobility transitions (Table 1). These are supplemented with an analysis of the stakeholder landscape in each of the cities included in the study.

**Table 1.** Number of interviews in each sector by specific city.

|  | Kigali | Nairobi | Kisumu | Dar es Salaam |
|---|---|---|---|---|
| Academia, NGOs, international organizations | - | 8 | 3 | 2 |
| Public sector—country level | 4 | | 1 | 1 |
| Public sector—city level | 1 | 1 | 4 | 3 |
| Start ups/private sector | 4 | 4 | 2 | 1 |
| Operators | 3 | 3 | 1 | 1 |

The content analysis encompasses the most important documentation, which was identified as relevant for potential e-mobility transitions based on its focus on environmental, energy or transportation sectors. The review was conducted at the level of country-, city- and program-level documents (see Table 2). The documents were screened for the occurrence of electric mobility/e-mobility/electric vehicles (EVs) terminology and, when present, analyzed in terms of the thematic context in which they were mentioned. After this initial screening, the sections identified as relevant for e-mobility transitions underwent a detail analysis, concentrating on identification of proposed solutions for the introduction and mainstreaming of e-mobility. In line with the most common themes presented in the documents, these contents were categorized into four distinctive themes, namely: (1) Incentives for e-mobility uptake; (2) Restrictive regulatory measures on conventional vehicles; (3) Incentives for integration of informal/semi-formal transport providers in e-mobility; (4) Project-level activities (although the information relevant for point no 4. was additionally supplemented with expertise of the authors in terms of recent on-the-ground initiatives present in the target countries and information gained through the interviews—see Table 3). The findings of the analysis are included in Section 4 of this article and organized according to the scope of the identified policies and documents (national/local) as well as the sectors they specifically concerned.

The interviews were conducted with relevant actors involved in or affected by e-mobility transitions in Dar es Salaam, Kigali, Kisumu and Nairobi. The recruitment of interviewees was conducted based on a purposive sampling method with an intention of reaching a multi-sectoral perspective on the issue in question. The interviews were conducted in English or a local language, recorded and transcribed and then underwent thematic analysis through coding of themes identified as relevant for this study. These themes were then gathered in a common template structured according to the represented sector and geographic origin of respondents. This allowed identification of commonalities and differences across specific contexts included in this study. Additionally, illustrative quotes, not only reflective of specific respondents' opinions, but also of the broader sectorial perceptions of discussed phenomena in local contexts, were identified and included in this article (predominantly in Section 6).

Overall, 47 interviews with representatives from the public sector, e-mobility start-ups, transport operators, academia, non-governmental organizations (NGOs) and international organizations were conducted between January 2020 and January 2021. A relatively equal distribution of sectors was attempted, although this was limited by the availability of respondents in specific contexts (see Table 1).

Consequently, rather than aiming for full representativeness, the study focuses on identifying perceptions on the transition process across different sectors as well as information on the anticipated socio-economic impact on operators. Those perceptions were juxtaposed with the existing policy environment.

**Table 2.** Policy environment of Kigali, Nairobi, Kisumu and Dar Es Salaam.

| | Kigali | Nairobi | Kisumu | Dar es Salaam |
|---|---|---|---|---|
| Country-level policies | — EV mitigation target in the updated Nationally Determined Contribution (NDC) (2020), following the third National Communication under the United Nations Framework Convention on Climate Change (2018)<br>1. Ongoing revision of the National Transport Policy, including EVs<br>2. Feasibility study on e-mobility in Rwanda (2019), upcoming World Bank electric last mile connectivity study<br>3. Ongoing work on standards | — National Climate Change Action Plan 2018–2022 announcing EV technical standards, incentives, pilot projects and public procurement, implementing the 2020 updated NDC supporting low-carbon and efficient transportation systems<br>1. Ongoing revision of the Integrated National Transport Policy (2009), including EVs<br>2. Target of 5% of imported electric cars annually by 2025 (National Energy Efficiency and Conservation Strategy)<br>3. Kenya Bureau of Standards (KEBS) 21 standards for EVs in 2019<br>4. Financial Bill of 2019 reducing excise duty rates for all battery electric vehicles<br>5. Exploratory work: e-mobility study, workshops, works on taxation, registration and importation of EVs | | — No transport policy mentioning EVs, but fuel switching in transport systems mentioned in its climate mitigation strategy (2012) |
| City-level policies | — Kigali Transport Master Plan 2050 (2020), with one unprecise reference to charging infrastructure<br>1. GGGI Quickscan charging infrastructure for Kigali e-buses (2020) | — Climate Change Action plan in the making mentioning e-mobility Consideration of one electric BRT corridor | — Kisumu Sustainable Urban Mobility Plan (SUMP) 2020 setting electrification targets | — Absent at this stage |
| Thematic focus | | | | |
| Incentives for e-mobility uptake | — Feasibility study recommendations: reduced import duties and VAT on EV parts, special electric tariff for charging stations<br>1. Custom tax exemptions granted by the Rwanda Development Board | — National incentives in the form of reduced excise tax for EVs from 20 to 10% in 2019; ongoing further work to reduce taxation and to facilitate importation and registration | | — Absent at this stage |
| Restrictive regulatory measures on conventional vehicles | — National level: targeted (upcoming) introduction of vehicle emission standards, including tax incentives inspection and scrapping older vehicles (2020 updated NDC) | — National level: age limit of 8 years to import second-hand cars<br>— City level: possibility for low-emission zones (2019 Nairobi City County Transport Bill) | — Absent at this stage | — Age limit of 10 years importing second-hand cars |
| Incentives for integration of transport providers in e-mobility | — Absent at this stage | — Absent at this stage | — Absent at this stage | — Absent at this stage |

**Table 3.** Project-level activities.

| Projects (Public and Private) | | | |
| --- | --- | --- | --- |
| **Kigali** | **Nairobi** | **Kisumu** | **Dar es Salaam** |
| − Projects driven by international and local actors at various levels of operationalization: e-cars (Volkswagen), e-motos (Ampersand, Safi Ride, Rwanda Electric Mobility), shared e-bicycles (Gura Ride) <br> − Participation in the e-mobility program SOLUTIONSplus <br> − Amerpsand's project on electrfication of moto-taxis with support from FONERWA <br> − Electric Last Mile Connectivity Study <br> − (World Bank) | − Projects led by foreign-led or Kenyan private companies: <br> − operational: e-car taxis (Nopea Ride), safari EVs (Opibus) <br> − small fleet: e-tricycle (Solar E-Cycle), deliveries e-light duty vehicles (Drive Electric) <br> − current pilots: e-motorcycles (Fika Mobility, Ecobodaa, Kiri EV Ltd., Stimaboda, WEEE center), e-tuk-tuks and handcarts (Auto Truck), wheelchairs (Lincell Technology), e-minibus and batteries (Opibus, Mazi) <br> − upcoming pilots of energy entities KenGen and KPLC <br> − upcoming research by Nairobi University (Mechanical Engineering Department) <br> − Public–private: upcoming pilot of 99 electric motorcycles to be distributed between Kisumu, Nairobi and Kisii, run by UNEP (GEF funding) in partnership with the Chinese company TAILG and the Kenya Power & Lighting Company | − Operational e-tuk-tuks <br> − Upcoming pilot on e-buses/matatus (Opibus) <br> − Exploration of future participation in SOLUTIONSplus | − Participation in the e-mobility program SOLUTIONSplus <br> − National academic institutions advancing research in e-mobility, e.g., Dar es Salaam Institute of Technology, the National Institute of Transportation (EV prototypes) |

## 3. Cities' Profiles—Kigali, Nairobi, Kisumu and Dar es Salaam

The four cities included in this study are dynamically developing commercial and cultural centers located in Eastern Africa. Nairobi, Dar es Salaam and Kigali are the biggest cities and urban agglomerations of their countries, inhabited, respectively, by 4.4 [20], 6.4 [21] and 1.29 [22] million people. Kisumu is the third biggest city in Kenya with an urban population estimated at 567,963 and a county population of 1.1 million inhabitants. While differing in size, geographic context and specific functions, they share commonalities in terms of the urban transformation processes they undergo. Additionally, the cities experienced stable to rapid population growth (Kisumu on average by 1.63% in the years 2000–2015 [23], Nairobi—3.8%, Dar Es Salaam—5.4%, and Kigali—4.2% on average in the years 2000–2018 [21]). A large proportion of this growth is concentrated in informal settlements, emerging in a peri-urban context in most of them. For instance, 62% of Kisumu's inhabitants are estimated to live in informal settlements. In Nairobi, the spatial structure inherited from colonial times reinforced socio-spatial and income-based segregation and resulted in dramatic divergences in residential densities [24] (half of its population on 5% of the total residential area only) [25]. In Kigali, intensive efforts to modernize infrastructure link with waves of evictions and the concentration of the low-income population in the peripheral "bedroom" types of districts [26]. Similarly, Dar es Salaam continues to experience a radical urban sprawl [27].

Some commonalities are to be observed in terms of the modal split, characterized by high shares of collective transport and walking, and of the evolution of urban transportation systems in the last decades. In Nairobi, walking, informal private minibuses acting de facto as public transport (*matatus*) and semi-formal buses are the dominant mobility options, with, respectively, 39.7%, 28.5% and 12.2% of the modal share in 2013 [28]. While Kisumu lacks a formal public transport system, moto-bodas (13.5%) and matatus/buses (13%) are, apart from walking (52.7%), the dominant form of transportation [29]. Similarly, in Dar es Salaam, 62% of trips happened with *daladalas* (minibuses), 17% with walking and 12% with private vehicles in 2014 [30]. In Kigali, the majority of trips are made on foot or

bike (52%) or by public transport (PT), mostly bus services (16%), but also motorcycle taxis (12%). The city is also characterized by low to moderate levels of motorization (approximately 15 automobiles for every 1000 inhabitants) [31]. Each of the cities initiated important investment into public transportation to various degrees, except for the case of Kisumu which is still planning to undertake a scoping study for a potential public transport project. Dar es Salaam is currently engaged in the Urban Transport Improvement Project with the support of the World Bank and the first phase of its bus rapid transit (BRT) system was launched in 2016. A number of measures are discussed or in the wake of being implemented in Nairobi in order to improve the mobility landscape. This includes the future introduction of BRT services—even though significantly delayed—renewal of railway infrastructure and vehicles and improvement of the non-motorized infrastructure, especially in the city core. Kigali has seen remarkable development of road and street infrastructure, and as part of this process, investments in safe sidewalks and incremental public transport reforms were advanced to benefit most of the urban trips.

While historically semi-formal transport operators played a relevant role in transport provision in each of the cities, currently this involvement varies across the region. The most advanced "formalization" process occurred in Kigali. The first reforms leading to reductions in push-taxis and moto-taxis were introduced in the city around 2005 [10]. Further public transport reform was initiated in 2008 by the City of Kigali Authority and led to the regularization and signature of bus operation contracts in 2013. Currently, motorcycle taxis provide a competing service to public transport buses on the same routes but, on the other hand, supplement public transport buses by providing door to door on-demand trips to various locations within the city. The City of Kigali government and traffic police department are engaging these motorcycle and bicycle taxi operators and an increased effort has been realized in the formalization process.

Public transport in Dar es Salaam is predominantly provided by a large fleet of privately owned minibuses (so-called daladala), which are operated informally with often non-organized schedules and route services [32]. In addition to these bus services, motorized two- and three-wheeler taxis (boda boda and bajaji, respectively) are very common. They are used by the population for shorter distances and enable feeder connectivity to the paratransit buses. In areas not served by buses, motorcycle taxis are the only publicly available mode of transportation and hence offer a de facto public transport service, filling a gap in the transport system. In the face of the mobility challenges facing the city, public authorities responsible for the sector have, in recent years, envisaged to phase out the minibuses on all major roads and replace them with bus rapid transit (BRT) in the medium to long term [33]. In the short term, current BRT initiatives introduced in the city in 2016 are expected to come along with the formalization of the existing private transport services and their integration into the entire public transport system in Dar es Salaam [34].

In Nairobi, the dominant modes of walking and collective transportation are undertaken in poor conditions including non-existent or poor-quality sidewalks, flawed bus services and, in the absence of urban planning for paratransit, public regulation being either inexistent or repressive [35]. Planning has been additionally criticized for its focus on large road transportation projects and its top-down approach, lacking civil society participation [21,36]. The public sector's efforts regarding boda bodas (moto-taxis) included top-down attempts to regulate them, unapplied to a large extent. Recent attempts to increase control over the transport sector included bus route permits, the theoretical possibility of low-emission zones [37] and controversial attempts to expel matatus from the city core.

Kisumu lacks a formal public transport system, and its role is played by privately owned *matatus* (minibuses). The matatu industry is a major employer and creates revenues for the public sector through the fees they pay to the County Government of Kisumu. Additionally, there is a rise in operations of two- (bicycles and motorcycles) and three-wheeler vehicles, which are often run by civilians seeking to make a living and yet form a crucial part of the mobility ecosystem in Kisumu. They supplement the matatus by

providing on-demand trips to various locations within the city as well as the rural outskirts of the city. The County Government of Kisumu is constantly engaging these informal transport operators and increased efforts have been realized in the current process to formulate a Kisumu Sustainable Urban Mobility Plan (SUMP) in partnership with the Institute for Transportation and Development Policy (ITDP) and the Ford Foundation.

## 4. E-mobility Policy Environment: Barriers and Opportunities for Different Mobility Providers

Electric mobility could provide opportunities for different types of mobility providers and stakeholder groups. However, this requires an enabling policy environment that allows participation in the transport system of public transport companies, corporations, associations, innovators and individual drivers alike. This subchapter reflects on these issues by scrutinizing the emerging policies on e-mobility across the studied countries and cities. Rwanda, Kenya and Tanzania stand at the beginning of a path towards policies on transport electrification. Analyzing the evolution of policies and project interventions over the last two years shows that electric mobility is increasingly identified by public institutions as a relevant low-carbon strategy, allowing mitigation of carbon emissions and a decrease in urban air pollution. Yet policies at national and local levels remain at a very nascent stage, fragmented and still uncoordinated, and in the case of Tanzania, they are mostly absent. Across the four analyzed East African cities, similar trends of a progressive elaboration—or external push for elaboration—of policies and regulations enabling electric mobility can be observed; however, the pace of adoption and levels of ambition vary in the four cases.

### 4.1. National Policies

Even still at a nascent stage, Kenyan and Rwandan authorities are engaged in the process of developing e-mobility strategies. The sequential logic of steps and policies (e.g., targets via climate change policies, transport policies, standards and financial incentives, feasibility studies), however, vary between the two countries. Kenya is more advanced in terms of standards and incentives but lacks a national mitigation target via e-mobility and a comprehensive feasibility study, two steps that Rwanda has already undertaken, before now working on incentives and standards. In Tanzania, no specific policies or standards are in place.

#### 4.1.1. Climate Change Policies

At the national level, climate change policy documents recently released mention electric vehicles. Rwanda was the first country in the region to submit its second Nationally Determined Contribution (NDC) in May 2020, where it identifies electric mobility as part of its climate change mitigation measures. EVs are expected to contribute to a reduction of 9% in GHG emissions in the energy sector in 2030 [38]. The NDC envisions a progressive adoption of electric buses, cars and motorcycles starting in 2020, replacing conventional vehicle sales and diminishing transport fuel imports. This measure is conditional, meaning that it depends on external financial support from donors such as development finance institutions (DFIs) and climate or private donors. Previously, e-mobility had been mentioned in the Third National Communication under the United Nations Framework Convention on Climate Change in the "Mitigation Scenarios and Reduced Emissions", but only for the adoption of electric cars, which planned to substitute 150,000 conventional cars by 2050 [39].

Kenyan authorities have not quantified a similar overarching mitigation target via e-mobility yet; however, the National Climate Change Action Plan 2018–2022 announced a series of measures facilitating the introduction of EVs and identified the opportunity to reduce 60% of two-wheeler emissions via a transition to electric motorcycles [40]. Studies have shown that e-mobility bears the second highest mitigation potential for transport emissions in Kenya [41] (as 86% of the electric generation mix is produced from energies qualified as renewable in 2018, mostly geothermal and hydropower) [42]. In addition,

the country has a substantial electricity surplus of 25–30%, raising a financial interest to see demand for electricity rising through transport electrification.

Tanzania here lags behind, as its NDCs and the National Climate Change Strategy lack explicit mention of electric mobility, even though the strategy outlines objectives and interventions in the field of energy, a crucial sector for e-mobility development, and promotes fuel switching in transport systems and low-emission transport via mass rapid transport systems [43]. Current significant developments in hydropower generation present opportunities in the transport, industry and manufacturing sectors, favoring increases in productive uses of electricity and e-mobility.

### 4.1.2. Transport Policies

Following the inclusion of electric mobility in climate-related planning documents in Kenya and Rwanda, national transport policies are slowly starting to integrate this technological transition.

The Kenyan draft Updated Integrated National Transport Policy (2020) includes a principle of governmental support toward the uptake of EVs, possibly going up to full electrification of land transport vehicles, and the need for further incentives and standards (safety of vehicles and personnel protection systems, charging infrastructure, infrastructure support). More recently, the Kenyan Ministry of Energy (2020)—not Transport—included a target of 5% EVs in total yearly car imports in its National Energy Efficiency and Conservation Strategy [44], raising the question of coordination between public institutions on e-mobility.

Rwanda is also in the process of revising its National Transport Policy, mirroring and operationalizing the climate mitigation target on e-mobility set in its updated NDC. This leans on findings from the 2019 e-mobility feasibility study commissioned by the Rwanda Green Fund (FONERWA), which kick-started the work towards a future e-mobility strategy. This study identified the possibility to reduce greenhouse gas (GHG) emissions by 17% in 2030 as compared to a business-as-usual scenario, provided EVs represent 30% of total motorcycles, 8% of cars, 20% of buses and 25% of taxis, minibuses and microbuses in 2030 [45]. Subsequently to the study, Rwanda President Paul Kagame announced in August 2019 his intention to replace conventional motorcycles with electric ones. In contrast, Tanzanian authorities have not revised their national transport policy yet to include e-mobility.

### 4.1.3. Standards and Incentives

Besides national transport policies, a regulatory environment is slowly emerging around technical standards and financial incentives, though at a different pace in the three countries.

Kenya is one step ahead: by June 2020, 21 technical standards had been adopted, covering vehicles, batteries and safety requirements. At the fiscal level, a 10-point reduction in excise duty for electric vehicles from 20% to 10% was approved in 2019. Current discussions touch upon a further reduction in excise duty, a targeted percentage of EVs within total imports by 2025 and modification of building codes to plan for the integration of charging infrastructure in public buildings and residential estates [44].

In Rwanda, the above-mentioned feasibility study positions financial incentives in the form of reduced import duties, value-added tax (VAT) exemptions or special electricity tariffs for charging stations, as well as technical standards and environmental standards (for instance, recycling of batteries) as crucial for the promotion of e-mobility in Rwanda and Kigali specifically. The Rwanda Standards Board is currently working on charging standards.

### 4.2. Local Policies

At a city level, e-mobility is a relatively new concept in local policies. At the time of data collection, local authorities were in the process of developing regulatory and facilitation strategies, not yet finalized or adopted. Again, analysis of the maturity of policies must

be undertaken with caution as the policy field is charging rapidly. Most local authorities lag, to some extent, behind national authorities, especially Kenyan and Rwandan ones.

The Kigali 2050 Transport Master Plan updated in 2020 entails one reference to electric mobility, namely, the deployment of charging stations at fuel stations in the city [46], yet without further details on the extent and location of deployment or the corresponding timeline. In Nairobi, local public institutions and partners are, since Spring 2020, in the process of drafting a local climate action plan, which may include measures supporting electric mobility. Such support will likely target projects addressing electric car taxis and electric motorcycles, both being the most mature initiatives among e-mobility developments. Impacts and barriers for these private projects to be executed will be examined, as well as supporting measures, through local policy.

The Kenyan city of Kisumu stands out with the draft Sustainable Urban Mobility Plan (SUMP) establishing targets for the integration of e-mobility in its transport system by 2030. This will start with a feasibility study for the electrification of public transportation in 2021–2022. The period between 2023 and 2030 is set out as an implementation period whereby the vision is to electrify 50% of Kisumu's motorcycle taxis and three-wheeler taxis by 2025 and fully electrify the bus fleet by 2030. In addition to this SUMP, Kisumu County has been involved in the United Nations Environment Programme (UNEP) demonstration project where electric motorcycle taxis will be deployed.

Finally, no local policy reference to electric vehicles could be found in Dar es Salaam plans. Despite this gap, Dar es Salaam authorities show their interest in electric mobility by taking part in the European Union (EU)-funded project SOLUTIONSplus, in which Kigali authorities also participate. The project encompasses demonstration actions which cover electric tuk-tuks (Dar es Salaam), e-shared bicycles, e-motorcycles and business models of e-buses (Kigali). Some thought is given to the electrification of public bus fleets on BRT lines.

At both national and local levels, the few nascent and sporadic electric mobility policies do not mention transport providers, involvement of them or impacts on transport systems. An exception is the feasibility report on e-mobility in Rwanda [45] which briefly mentions the need to include transport operators in the transitioning process as they may be under the threat of job loss.

## 5. Electric Mobility Stakeholder Landscape in Kigali, Nairobi, Kisumu and Dar es Salaam

The literature shows that transportation is generally considered as a system and a functional sector which requires a wide range of institutions to support it [47]. Such multiplicity of institutions calls for greater importance to be placed on the roles of actors and players in the sector as well as the relationships and networks that occur between them [48]. In the four cities under study here, it is evident that several actors and stakeholders, both public (ranging from national to local) and private, including new e-mobility companies and traditional transport providers, are involved in the process of introducing EVs and the corresponding infrastructure, or will be impacted by this transition. Yet the intensity of involvement strongly varies among these stakeholders. New e-mobility players drive projects on the ground (Table 3), triggering thoughts from public authorities on policies enabling e-mobility. The involvement of traditional transport providers is limited or variable, mostly depending on the strategies of the new e-mobility companies.

### *5.1. Public Authorities*

Overarching transport policy development in all four cities is the responsibility of national-level institutions. Yet coordination of urban transport infrastructure and services is a challenging task, especially in Kenya where 15 public agencies are responsible for urban transport, with overlapping duties [49]. In addition to these urban institutional challenges, e-mobility adds a further complexity layer as it potentially mobilizes public authorities covering transport, energy, industry, environment, research and development segments. In the study countries, not only mobility institutions and parastatals are getting

active, but also energy stakeholders and industry representatives. For instance, in Kenya, energy actors are looking for vectors to increase electricity consumption in the context of oversupply, explaining the involvement of the Ministry of Energy on targets for the share of EVs in total vehicle imports, and of the Energy and Petroleum Regulatory Authority (EPRA) in feasibility assessments and workshops.

At the local level, the responsibility for the overall development of cities is generally entrusted to the local city authorities as it is in the case of all four cities. The Constitution of Kenya assigns fourteen functions to the county governments in Kenya, one of them being county transport planning, including construction and maintenance of county roads, street lighting, traffic management, parking and public road transport. The Dar es Salaam City Council is the decentralized local government institution in charge of coordinating developmental issues cutting across the five municipalities (Kinondoni, Ilala, Ubungo, Kigamboni and Temeke) under its jurisdiction [50]. The five municipalities are responsible for the provision of basic social services including urban planning. Likewise, the City of Kigali government takes full responsibilities for the development of areas under its respective jurisdiction. The case of Nairobi is specific, with the transfer of the county transport responsibilities to the national level and the creation of the Nairobi Metropolitan Services (NMS) office, placed under the president's Executive Office as of June 2020. In addition, urban governance structures have been characterized by the creation of entities coordinating transport at the local level in Dar es Salaam, Nairobi and Kigali, touted as a way of reducing institutional complexities, especially in cases where the innovation deployment cuts through several local administrative boundaries [51]. Such new entities include the Dar Rapid Transit (DART) Agency, the Nairobi Metropolitan Area Transport Authority (NaMATA) and the Rwanda Utilities and Regulatory Authority (RURA), playing an active role in the regulation of urban transport services in Kigali.

Though it is expected that reform processes happening at the local level include both national and local public authorities, it is found that national authorities are often cited as dominating transport evolutions, including e-mobility, even those happening at the local level. Exploratory studies, discussions, workshops and e-mobility have been mostly led and held at the national level in Rwanda and Kenya. As mentioned above, except for the case of the Kisumu SUMP, thoughts on national policies on e-mobility, even though at the stage of drafts, are more advanced than local policies. Hence, in Nairobi, for instance, national authorities have until recently been perceived as dominating the e-mobility sphere, with little to no involvement of the county's administration, even in the case of the motorcycle pilot to be deployed in Nairobi and Kisumu. National project leaders were seen as "people from the Ministry coming to run a project at the County level without ownership at the County level" [52]. This limited involvement of the county must be framed within the past context of the political leadership crisis in Nairobi, resulting in the transfer of the responsibility and creation of NMS.

*5.2. New E-Mobility Private Players*

Transport authorities in all study countries have policies and plans that place emphasis on private sector involvement. It is therefore not surprising to note that there is a gradual rise in new private sector mobility initiatives in all four cities. An exploratory review of technology-enabled mobility start-ups in Africa (from 2010 to 2019) revealed that Nairobi, for instance, is among one of the cities with the highest concentration of mobility start-ups in Africa, involved in shared mobility services, technology innovations for vehicle performance, commuter experience or data-driven decision making [53]. Kigali, Dar es Salaam and Kisumu are also experiencing this emerging rise in start-up activities.

Regarding electric mobility specifically, projects already carried out on the ground by private actors and start-ups, mostly in Kenya and Rwanda, play a critical role in the e-mobility discussion. In Kenya, the range of EV types explored by private e-mobility companies is significantly broad. Several projects are led in Nairobi, introducing or still exploring e-car taxis, e-safari vehicles, e-motorbikes (vehicles and charging), e-tricycles

and e-tuk-tuks, e-light duty deliveries, e-handcarts, e-wheelchairs and, in the upcoming months, e-minibuses. Similarly, pilots are led in Rwanda by foreign and local companies such as Volkswagen, Ampersand, Safi Ride, Rwanda Electric Mobility (REM) and Gura Ride. Some electric tuk-tuks are also operational in Kisumu. Dar es Salaam differs, currently not showing such a dynamic e-mobility pilot landscape in urban settings, but rather in rural ones, or at the research stage.

As e-mobility companies are driving pilot projects on the ground in Rwanda and Kenya, they are tightly embedded in discussions with public institutions on upcoming policies and regulations shaping the uptake of e-mobility. Such talks are either organized by public institutions themselves (e.g., multi-stakeholder workshops hosted by the Ministry of Infrastructure in Kigali in February 2020) or attended by civil servants when organized by third parties such as industry associations, supranational entities or DFIs (e.g., November 2019 conference on e-mobility in Kisumu organized by Siemens Stiftung, UNEP and the German development agency (GIZ); February 2020 conference in Nairobi hosted by the Kenyan Ministry of Energy partnering with the Association of Energy Professionals Eastern Africa as well as the German development agency (GIZ) workshops in 2019 and 2020). In these fora, start-ups and private companies are vocal about incentives that they consider necessary to scale up their pilot projects. The market-led development is clearly expressed in the 2019 feasibility study in Rwanda, stating that the approach should be "inclusive and rely on market actor initiatives and innovation" (p. XII), while "government initiatives should support but not distort the market" (p. XIII).

Contrasting with this private dynamism in three of the four cities, publicly led projects have been delayed. For instance, pilot projects in Kenya testing electric motorcycles and tuk-tuks in two counties by 2020 and public procurement of 150 electric hybrid buses and cars by 2019 (unspecified distribution by vehicle type) were announced in the 2018–2022 Climate Change Action Plan [54], but these were not operational as of November 2020 with the COVID-19 pandemic further delaying the implementation process.

### 5.3. Traditional Transport Providers

There is a strong presence of informal transport operators mainly made up of mini-bus operators, moto-taxi and three-wheeler operators in all four cities. In all the cities except in Kigali, where non-licensed minibuses are being phased out [55], private minibus operators continue to provide the greater share of motorized public transport services as is the case in many African countries [56]. There is also growing modal shares of moto-taxi services which account for 12% in Kigali [46] and 13.5% in Kisumu [26]. In Nairobi, two-wheelers still accounted for a moderate modal share of 5.4% in 2013, but their progression was one of the most dynamic between 2004 and 2013 in the city [28] and at the national level [57]. A similar growth phenomenon is prevalent in Dar es Salaam as well [58].

Despite this strong presence, representatives of transport providers (associations) seemed limitedly involved in discussions with public authorities on electric mobility at the time of data collection. In Nairobi, for example, the local administration has seemingly given priority to start-ups, seen as a "higher" or "strategic" level, contrasting to a "lower" level of the transport representatives [59]. The Boda Boda Safety Association of Kenya (BAK) and local authorities do not seem to be in discussions on electric mobility, which the former regrets. In addition, although the BAK is formally on the list of involved partners in the national motorcycle taxi pilot, interviews revealed a limited role conferred to the BAK. The same seems to apply with regard to minibus transport workers, as their representative Transport Workers Union (TWU) states not to be in talks yet with public institutions or start-ups on the topic of e-mobility [60].

With regard to relationships between start-ups and motorcycle taxi drivers directly and their representatives, patterns seem to fluctuate, mostly depending on the strategies of e-mobility companies. In Nairobi, two start-ups took the initiative to engage with the BAK, which resulted in the involvement of the latter to facilitate contacts between these two companies and individual drivers, including interviews and data collection in the

perspective of market entry. Apart from these two private players, the BAK does not seem to appear as a natural or priority partner to other start-ups. Engagement also varies as representative associations differ (BAK or another association named Sauti ya bodaboda) depending on the local area [61]. In contrast, delivery companies using motorcycles for various services, including food products, are commonly seen as a low-hanging fruit. Direct contacts between some e-mobility companies and motorcycle drivers happen in Nairobi and Kigali—depending on the companies' strategies, mostly in order to collect data on internal combustion engine vehicles and test the feasibility of and the transition to the new vehicles.

It is important that the discussion on the involvement of transport providers in e-mobility is gradually shaped in the larger context of stakeholder involvement in mobility policies. In Dar es Salaam, where similar limitations on involvement of informal operators persisted over the years, local authorities had taken note and were making efforts to create active engagement platforms for all stakeholders. A current Dar es Salaam Urban Transport Improvement Project to expand the BRT development in Dar es Salaam places emphasis on the need to actively engage the existing informal transport operators in the city. The project for that matter has dedicated a specific sub-component to support the existing daladala (minibus) operators to establish companies, cooperatives or franchises in line with the sector transformation efforts that will see the hitherto informal operators become some of the licensed operators of future BRT phases [10].

### 5.4. International Organizations, Development Finance Institutions and Academia

Academia and research institutions, both national and foreign, also play essential roles in the co-development of e-mobility in the study cities, where there is growing research activities into the development of electric mobility products and services, and in assessing the baseline scenario of internal combustion engines (e.g., fleet assessment).

Finally, NGOs (e.g., Transport for Africa in Kenya) and international organizations such as UN Environment or UN-Habitat play an important supporting role. In Kenya, the "Socially Just Public Transport Working group", a hybrid thinktank organized by the German Foundation Friedrich Ebert Stiftung, works on principles for just transportation, including from the perspective of new mobility options. Other examples of stakeholders playing an important role in the e-mobility landscape include actors such as the World Bank, DFID, GIZ and several private sector foundations.

### 6. Opportunities and Barriers for Promotion of E-Mobility in the Local Transportation Systems—Perceptions on Transitioning Process

The analysis of the interviewees' perceptions of incentives and barriers for the mainstreaming of e-mobility presents quite a complicated and diversified picture across the reviewed sectors and cities. However, there are some common traits, which can be identified across the region, particularly in the context of the incentives which drive the development of e-mobility.

Firstly, in reference to the ecological gains framed more generally as climate change mitigation potential or concretely as reduction in air pollution, green city development is strong. Those factors were relevant for public sector representatives in Tanzania and across all analyzed sectors in Kigali and Nairobi. While the narrative of tackling climate change was particularly prominent on the side of the public sector, international organizations and academia, operators in Kigali and Nairobi also mention this as relevant. Similarly, the reduction in dependency on petrol and the shift to renewable energy sources produced at a national level were underscored as relevant for representatives of various sectors in these three cities. Alongside environmental gains, a significant proportion of representatives from start-ups, transport providers, international organizations and local authorities, particularly strong in Kigali, Nairobi and Kisumu, expect that the shift to e-mobility, while challenging, may reduce the operational and maintenance costs of vehicles. This is especially the case for smaller vehicles (electric two- and three-wheelers). These environmental and socio-economic expectations explain a high enthusiasm captured in Nairobi,

where mobility electrification is regarded as a disruptive technology, qualified as a "godsend" [62] and a "huge game changer" [63,64]. Otherwise, a patchwork of secondary heterogenous opportunities is diversely cited, including opportunities for job creation, active start-up scenes and the private sector, absence of noise, renewal of urban planning practices and a global drive for e-mobility, including international funding and support.

Yet the range of barriers remains high. Contrary to incentives, more commonalities can be identified across cities included in the study. First and foremost, some negative perceptions on EVs and a lack of awareness concerning the positive aspects of e-mobility are considered as major issues, mainly from a public sector perspective. Representatives of the sector tend to consider transport operators and the general public as the ones who might be ill-informed in that regard. This is not universally confirmed on the side of operators who tend to see both challenges and potential opportunities linked with the transitioning process. As underscored by a start-up representative, "if people are not convinced that this product can work they will not come to buy your product. In the end you end with a huge investment which will not really materialize" [65]. Similarly, operators point out that "we also have to see the advantage in using them, compared to what we have been using before. If there is cost effectiveness (...) we have to make a comparison to see what can come out of it" [66].

Linking with this, significant attention in Kigali, Dar es Salaam and Nairobi was given to the issues related to bad road infrastructure and the inaccessibility of specific areas of these cities. As pointed out by one of respondents in Kigali, "engine-motor can go everywhere in the country, even in the countryside, even on the bad roads but you have to compare now with this one (...) our drivers are not riding only on these paved roads they go also in this tiny roads, and they go even to very, very bad roads, so we have to do this survey if it can be very effective" [67]. The ability to drive electric vehicles under similar conditions of challenging road infrastructure as well as with heavy weights transported (freight, passenger) was also questioned in Nairobi. Similar weight was given to potential issues related to the costs of deployment of a sufficient e-mobility charging infrastructure (charging and swapping stations), which was particularly strong in Kigali and in Nairobi. In Dar es Salaam, the issue of the lack of land availability was only mentioned in the context of informal settlements, while the development of stations, in itself, was seen as an opportunity for the private sector. Overall, the urban planning procedures regarding these stations seem inexistent or unclear, representing a planning void in which start-ups in Nairobi and Kigali navigate also by arranging those facilities by themselves. As mentioned by one start-up representative in Nairobi, "I think for now it's too early and everyone is doing whatever they think is right ( . . . ) there is no regulation around it and it's something that hasn't come up yet and considering how young the concept is. No one has thought about that because it's something that's going to crop up maybe in 3, 4 years from now" [68]. While some discussions were taking place, for instance, in Kigali, in terms of allocation of land for stations, representatives of multiple sectors point out the insufficient development in this field.

Lastly, some more attention was given to the issues of electricity reliability, costs and access in Kigali and Nairobi as well as missing policies. On top of those more universal problems, some local and sectoral issues were mentioned in different contexts. Kisumu is a particularly strong example here. While it is unlikely that issues of availability of charging stations are absent, or that there is no concern of high upfront investment for operators, the respondents focused mostly on the issues related strongly to the "secondary" status of the city, namely: availability of funding, the political process and the distance from the capital.

When analyzing the interrelationship between different sectors and their role in promoting e-mobility, a relatively clear picture can be drawn. In spite of quite early stages of policy formulation, public authorities at a national level identify the development of incentives and the creation of a conducive environment in terms of formulation of the legal and regulatory framework as their natural role. However, this self-

positioning concerns a broader shift of the policy environment to sustainability pathways, rather than championing e-mobility specifically. They position the private sector as their main counterpart, though this category includes mostly business people and the start-up scene. In fact, the latter may be considered as the key drivers of change who put pressure on public sector stakeholders in various arenas of advancement for the mainstreaming of e-mobility (perhaps with lower influence in Dar es Salaam where the pace of the start-ups in the sector is still slow). While they eagerly engage in partnerships with city and governmental counterparts, they remain critical regarding some aspects of the existing institutional frameworks. As mentioned by one of the start-up representatives in Tanzania, the issue of limited capacity and lack of technical support provided to e-mobility start-ups is noticeable. Even in Kigali where the government pushes for various eco-friendly policies, the start-up representatives notice that some policies are missing and there are some bureaucratic barriers which need to be overcome. In spite of these issues, an active cooperation is flourishing in different constellations of actors, particularly between international NGOs, academia, industry and start-ups, for instance, within international programs promoting e-mobility such as the EU-funded project SOLUTIONSplus, which aims to facilitate the co-development of innovative e-mobility solutions among local and international actors.

The role and positioning of operators, paratransit and individual "informal" actors are more ambiguous. At the current stage, involving those in e-mobility transitions has not been positioned as a priority. For instance, in Kigali where the policy environment is the most advanced, the public representatives mention that "we have not yet engaged them in a sufficient way, but we do it little by little because it requires some conducive environment, we have been working on policies" [69]. Similarly, the necessity of high upfront investment is acknowledged. Challenges linked to transitioning, especially in linkage to the coronavirus pandemic, are mentioned: "with COVID-19 there is a tremendous amount of stress that has been put on the resources, that means busses and other things" [70]. Occasionally, as in the case of Kisumu, public sector representatives believe that operators also feel included in better communication with the government. Conversely, they also believe the operators need to transform their organizational model to be ready to operate effectively. This perception is shared by one of the representatives of the public sector in Kigali: "the other issue is bus operators sometimes claiming they are not getting their benefit (..). But of course we are trying to see how we can assist them not only incentivizing them in terms of money but giving them some non physical interventions to help them out to get (...) their benefits but also improve their way of doing business" [71].

The ambiguity in the positioning of transport providers is also reflected in the opinions of the sector's representatives about e-mobility. High upfront costs for transitioning to electric vehicles are well acknowledged. Balancing higher upfront investment costs with lower operational and maintenance costs is more frequently mentioned for smaller vehicles (comparatively smaller investments resulting from a smaller battery size), especially when drivers are themselves owners and when combined with innovative business models (vehicle lease-to-own, rental of the battery, pay-as-you-go schemes). For buses or minibuses, this could be more challenging considering higher investment costs and the decoupling between owners and drivers responsible for fuel expenses in Kenya, for instance. This ambiguity is captured well by a bajaji (three-wheelers) sector representative in Dar es Salaam, who sees both potential for decreased costs but also competition between various actors involved in the process: "I see this competition to raise even higher because I think operational costs for electric bajaji's will be lower, which will in turn lower the transport charges for users leading to increased competition, not just with the taxis but with the conventional bajajis" [72].

In addition, a fear of job losses exists in the matatu segment in Kenya, strongly emphasized by the interviewee representing transport workers (TWU interview, May 2020). This was explicitly framed within current tensions around BRT implementation. Electric buses were seen as a threat to and replacement of current matatus and their culture, linked with a reduction in the number of workers. The interviewee was not

aware of current projects to convert existing buses with an electric powertrain (retrofit), instead of bringing in new electric buses. Dynamics of e-mobility acceptance (as well as implementation and cost characteristics) therefore appear as differing between modes.

Finally, some hypothetical benefits are "mode-specific", in the sense that they relate to current challenges faced by these modes. For instance, in Nairobi, a cashless payment in the minibus segment is envisioned by transport workers in the wake of electrification, while e-mobility projects could possibly improve boda boda road safety records via slower EVs or driving training programs.

## 7. Discussion and Conclusions

The conducted policy and project documentation analysis reveals that the three East African countries studied have initiated steps of a transition towards transport electrification, yet at different paces and taking diverging approaches in terms of sequential order (targets, policies, standards, incentives). Overall, they are still at an early stage of policy formulation, and detailed policies are still in the making. More advanced policies are in place in Rwandan and Kenyan contexts where electric vehicles are mentioned, particularly in various documents concentrating on environmental and climate policies, including documents such as the NDC or Climate Change Action Plan. There are some technical standards on e-vehicles recently introduced in Kenya and under development in Rwanda. Similarly, in Rwanda, a more strategic approach concerning detailed mainstreaming methods of e-mobility is included in a comprehensive feasibility study. Relatively weaker attention is given to e-mobility in city-level policies, which, except in Kisumu, only vaguely mention the phenomenon or are currently working towards eligible strategies.

The formulation of city-level policies and strategies is less advanced in spite of dynamic, on-the-ground developments. At the time of data collection, projects, mostly privately led by small companies, were burgeoning in Kigali, Nairobi and Kisumu, albeit with limited fleets or at stages of prototyping and piloting. The Tanzanian case stands out as least advanced in the field of e-mobility at a project level, in Dar es Salaam, though e-mobility projects are happening in rural areas, as well as at a policy level. These findings contribute to the nascent discussion on e-mobility developments in East Africa, where only a limited number of policy reports or students' theses have been published so far. These focus on the technical and financial feasibility, overall sustainability and environmental and economic impacts of EVs [73–78]. A gap remains in electric buses, since most studies cover the most mature EVs, namely, electric two- and three-wheelers.

The second question of this study, concerning the role of different stakeholders in the studied countries, reveals that greater on-the-ground mobilization of various sectors is visible than the policy environment would suggest, particularly in Rwanda and Kenya. As the analysis of incentives and barriers revealed, a growing number of private stakeholders are looking at meeting the decarbonization needs identified by public authorities, international organizations, donors and NGOs, combined with air pollution reduction, while exploring potential benefits from the lower operational and maintenance costs of EVs. Yet a set of financial and technical barriers persists as high upfront investment costs in vehicles and infrastructure constraining the uptake of such private initiatives, combined with persisting interrogations on the capacity of EVs to provide similar uses, compared to current fossil fuel vehicles in terms of road conditions, carried weight, range and speeds. These findings on environmental and economic drivers, as well as financial and technical hurdles for transport electrification, are consistent with other studies on e-mobility in low- and middle-income countries [79], where affordability, total cost of ownership and performance are significant aspects [11,80,81]. Regardless of these barriers, start-ups remain the most active actors in the field of e-mobility transitions, quite universally across the urban contexts (with the exception of Tanzania, where start-ups are more active in rural areas). There is also a recognizable effort between start-ups and public sector officials to build coalitions towards mainstreaming of e-mobility. For the former, these coalitions are crucial as they may allow them to remove obstacles and procedural difficulties they encounter along the way and

may open various business opportunities. The latter are incentivized by the fact that e-mobility can support them in reaching environmental goals defined within the policies they support, alongside economic goals (manufacturing, energy independence, economic growth) fitting in a perspective of "low-carbon development" [82]. Finally, our study shows that conventional operators, who might have an equally important role in the transitioning process, are still relatively marginalized in the formation of interest communities.

This leads to the third question of this study concerning the positioning of informal actors and operators in e-mobility transitions. As an emerging phenomenon with little maturity, this problem requires further analysis over time. While there is a significant body of literature focusing on economic hurdles that operators might experience in relation to the process of the "formalization" of their activities, up until recently, this topic was weakly discussed within the sphere of e-mobility. Beyond the East African case, the economic impacts of a transition to electric vehicles on transport drivers have started to be analyzed in Asian countries where e-mobility is more advanced (for instance, in the Philippines [83,84], India [85,86] or Bangladesh [87]). Nonetheless, the question of their involvement in policy formulation, planning and the recognition of their needs in these transitions in the Global South is still under-researched. This study found out that operators and their representative associations are limitedly recognized as major players in the transition, far behind new e-mobility players (start-ups) and public authorities, mostly national ones. Similarly, in the nascent policy sphere on electric mobility, the role of the sector is not, or not truly, recognized. While public authorities and start-ups are exchanging on policies, technical feasibility and incentives, transport operators and their representatives stated not to be truly involved in e-mobility policy talks with public authorities yet. Some start-ups entered in contact with drivers and representative associations in order to collect data on transport patterns, especially on the two-wheeler segment, but not all of them. In light of the above, the capacity of the initiated transitions to fully account for the needs of transport providers remains an open question. This is especially the case for buses and minibuses which provide mobility services for a considerable share of urban dwellers in the four cities of the study and where high investment costs will be needed to shift to electric mobility. Early tensions could be felt with the transport union in Kenya, interpreting electric mobility transitions in the light of current issues encountered with the BRT system implementation. Overall, a challenge of constrained financial resources is identified in the context of the COVID-19 pandemic. The question of acceptability versus potential future oppositions and events of public contestation in the like of the modernization program for jeepneys in the Philippines [88] remains open.

Hence, this analysis of the early stages of electric mobility developments and current (subject to rapid change) governance traits leads us to identify three gaps which should be adequately addressed by policymakers and revolving organizations (international, non-government, academia).

Firstly, this analysis identifies the need for significant further research to lead and call for continuous study of electric mobility in the region, looking at it at a more granular level and distinguishing between transport modes and vehicles. As discussed, effects of electrification seem likely to vary between vehicle types as investment costs vary, linked with the size of the battery and the vehicle, as well as technical characteristics. The additional layer of electrification comes on top of already different operational patterns, making the case for dissociated analysis.

Secondly, looking at current loopholes in the involvement of stakeholders in electric mobility, a principle of transparency of decisions and policy formulation should be set, allowing for structured engagement from the transport sector. Consultation of transport providers should be the minimal step, while public authorities ought to aim for co-production schemes enabling taking on board needs and expectations from transport providers. This is paramount for the success of electric mobility transitions, especially when identifying organizational and financial models which could ease the transition and alleviate investment constraints for transport providers (for instance, leasing or rental

models, pay-as-you-go). The potential for progressive solutions where transport providers play a crucial role may be higher than expected by the main gatekeepers facilitating the electric mobility transitions. However, from public sector perspectives, cooperating with on-the-ground players may seem risky not only because of their expected resistance but, in some contexts, also their engagement in politicized arenas or perception of being messy and uncontrollable. Bridging this divide emerges as a crucial step in truly mainstreaming the positive role of electric mobility both in transportation and climate change mitigation but also in an inclusive transitioning process.

Thirdly, urban planning is a further dimension currently omitted in e-mobility as revealed during interviews, creating uncertainties and leading to *bricolage* practices from stakeholders. Yet this has significant impacts, for instance, on availability—location of charging infrastructure, safety of location at fuel stations, etc. Similarly, on-the-ground actors, such as operators and start-ups, see an urgent need to demonstrate that electric vehicles can operate in weakly developed parts of local cities. In addition, this raises the question of the impact of electrification on transport systems, as neglecting urban planning may result in rebound effects and missed opportunities, such as the potential improvement of connectivity between transport modes via well-designed electric charging spots. Being limited to the absent involvement of transport providers further misses the opportunity to leverage electric mobility transitions to review planning practices, mostly ignoring informal to semi-formal transport services and infrastructure in planning, with the exception of Kigali. The possibility to integrate transit stops and waiting points into planning via the transition to electric mobility is not informed or not much discussed at this point.

In conclusion, this research calls for stakeholders to look at implementing *inclusive* electric mobility, based on modalities that include transport providers, potentially including redistributive effects benefiting them, and address their financial constraints.

**Author Contributions:** Conceptualization J.G., O.L.; methodology, J.G.; formal analysis, J.G., E.M.; investigation, E.M., A.N., J.A.O., J.S., J.G., E.T.; data curation, J.G., E.M., A.N., J.A.O., J.S.; writing—original draft preparation, J.G.; E.M.; A.N.; J.A.O.; J.S.; E.T.; writing—review and editing, J.G.; E.M.; A.N.; J.A.O.; supervision, J.G.; E.M.; project administration, O.L.; funding acquisition, O.L. All authors have read and agreed to the published version of the manuscript.

**Funding:** The preparation of the article was supported by the SOLUTIONSplus project, which has received funding from the European Union's Horizon 2020 research and innovation program under grant agreement No 875041 as well as UN-Habitat through the Urban Change Makers program.

**Institutional Review Board Statement:** The study was conducted according to the H2020 ethics standards, and approved by the SOLUTIONSplus consortium (January 2020).

**Informed Consent Statement:** Informed consent was obtained from all subjects involved in the study.

**Data Availability Statement:** The data presented in this study are available on request from the corresponding authors. The data are not publicly available due to confidentiality agreement with the interviewees included in this study.

**Acknowledgments:** We would like to thank all of the interviewees who agreed to share their knowledge and take part in this study.

**Conflicts of Interest:** The authors of the article are involved in the SOLUTIONSplus project, dealing with the roll-out of e-mobility solutions in Rwanda and Tanzania.

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
