# Peer review of "East Africa’s Policy and Stakeholder Integration of Informal Operators in Electric Mobility Transitions—Kigali, Nairobi, Kisumu and Dar es Salaam"

_sustainability, doi:10.3390/su13041703_

Round 1

Reviewer 1 Report

This paper is well-written with interesting discussions for future e-mobility development strategies and policies in East Africa.

Author Response

Thank you for your positive feedback

Reviewer 2 Report

  • Specify just from the beginning that "e-mobility" is used in the text  as electric mobility;
  • Specify in the main text the aim of the study;
  • Add the research questions;
  • Describe  the process applied to analyze the qualitative data; describe the content analysis process (the steps followed, according to this research method); describe the interview analysis process (was a thematic content analysis used? what were the steps followed to analyzing the data collected?);
  • Discuss the research results according to the research questions;
  • Discuss the research results in the light of other studies/research; are (or not) the own research results supporting the conclusions of other studies?;
  • Don't use an abbreviations without explaining it before (see, for e.g., the SDGs );
  • Use in the text the citation style recommended by the journal, please consult Sustainability | Instructions for Authors (mdpi.com)

Author Response

Thank you for your comments. Please find the answers directly next to the concrete remarks 

  • Specify just from the beginning that "e-mobility" is used in the text  as electric mobility;

    This has been clarified in the abstract. 
  • Specify in the main text the aim of the study;

    The main aim of the study was specified on the page no 3:
    'The aim of the study is to investigate the policy level solutions and stakeholder constellations established in the context of e-mobility in East Africa and to specifically reflect on the positioning of informal operators in the e-moblity transitions'
  • Add the research questions;
    We included research questions on the page no 3: 
    RQ1. How the country and city level policies in the selected contexts tackle e-mobility transitions?
    RQ2. Which stakeholders take an active role in mainstreaming of e-mobility and what roles are those?
    RQ3. What is the positioning of informal/semi-formal operators in the e-mobility transitions?
  • Describe  the process applied to analyze the qualitative data; describe the content analysis process (the steps followed, according to this research method); describe the interview analysis process (was a thematic content analysis used? what were the steps followed to analyzing the data collected?);

    This was included in the section Materials and methods, page no 4. Yes the thematic content analysis was used, we specify now in detail all of the steps taken during the data analysis phase. 

  • Discuss the research results according to the research questions;

    Overall specific sections of the article focus on specific questions. Section 4 predominantly covers RQ1, Section 5 - RQ2, Section 6 - both RQ2 and RQ3. 
    The final section of this article was reworked in order to discuss the findings according to the research questions.
    We kept the second part of discussion/conclusions section as it identifies gaps and includes recommendations for further action and research. 
  • Discuss the research results in the light of other studies/research; are (or not) the own research results supporting the conclusions of other studies?;

    This was included in the final section. We took a focused approach here and concentrated on literature concerning on electric mobility transitions specifically. 
  • Don't use an abbreviations without explaining it before (see, for e.g., the SDGs );

    This was corrected through the whole text. 
  • Use in the text the citation style recommended by the journal, please consult Sustainability | Instructions for Authors (mdpi.com)

    The references section was updated. We were unable to create brackets around references located directly in the text. In case of the acceptance of the paper we would hope this can be easily fixed during the formatting/corrections stage. 

Additionally we made a couple of small updates beyond the suggested comments (mainly including the title and the abstract). 

With best regards,

Authors 

Round 2

Reviewer 2 Report

The authors made the recommended improvements but there still is a problem with the citing system of the references in the text of the paper, please check the requirements of the journal related to this aspect. Also, check how are cited the references no. 10 (See reference 10..), no. 26 ( See reference 23...), no. 35 (See reference no. 10...???),  no. 41 See reference no. 26...???), no. 64...no. 68  in the List of references.

A separate file with specifications on how the authors have addressed each improvement according to each recommendations would have been very useful to reviewers (as usual is practiced). 

Author Response

Thank you for additional comments. The citing system was adapted to the journal's style, including interviews which are now cited through endnotes rather than directly in the text. Finally the specific references mentioned in the review were updated.

These changes appear through the whole text wherever we referred to literature or interviews and are marked in track-changes mode. 

Apologies for not including a separate file in the previous review round. We mentioned the main changes in the letter, however in most of the cases without a reference to specific pages. We include here a very short specification of the main changes from previous round (also marked in track changes in the previous file) 

research questions - p. 3

methods - p. 4

reformatted research results - p. 17 & 18

linkage of results with other studies p. 17 & 18